# Pipeline Leakage Detection Based on Secondary Phase Transform Cross-Correlation

**DOI:** 10.3390/s23031572

**Published:** 2023-02-01

**Authors:** Hetao Liang, Yan Gao, Haibin Li, Siyuan Huang, Minghui Chen, Baomin Wang

**Affiliations:** 1Huaneng Hunan Yueyang Power Generation Co., Ltd., Yueyang 414002, China; 2Institute of Acoustics, Chinese Academy of Sciences, Beijing 100190, China; 3Huaneng Clean Energy Research Institute, China Huaneng Group Co., Ltd., Beijing 102209, China

**Keywords:** acoustic emission, leak detection, cross-correlation, phase transform, time delay estimation

## Abstract

Leaks from pipes and valves are a reputational issue in industry. Maintenance of pipeline integrity is becoming a growing challenge due to the serious socioeconomic consequences. This paper presents a secondary phase transform (PHAT) cross-correlation method to improve the performance of the acoustic methods based on cross-correlation for pipeline leakage detection. Acoustic emission signals generated by pipe leakage are first captured by the sensors at different locations, and are subsequently analyzed using the cross-correlation curve to determine whether leakage is occurring. When leakage occurs, time delay estimation (TDE) is further carried out by peak search in the cross-correlation curve between the two sensor signals. In the analysis, the proposed method calculates the secondary cross-correlation function before the PHAT operation. A sinc interpolation method is then introduced for automatic searching the peak value of the cross-correlation curve. Numerical simulations and experimental results confirm the improved performance of the proposed method for noise suppression and accurate TDE compared to the basic cross-correlation method, which may be beneficial in engineering applications.

## 1. Introduction

Maintaining the reliability and integrity of pipeline networks is becoming an increasingly critical challenge in industry. A large number of pipes and valves exist in power plants for the transport of liquids and gases, and delays in detecting and repairing the damaged pipe sections can result in significant financial loss and hazardous situations. In long-term service, the pipeline networks are constantly affected by high temperature, high pressure, corrosion and damage, all of which makes them susceptible to leakage. Leakage detection is crucially important for a company when making asset management decisions and maintaining the resilience of the pipeline system. In recent decades, much attention has been paid to targeting leakage effectively and efficiently in academic and industrial communities [1,2,3].

Traditionally, leak detection surveys are conducted manually, for example, by listening to the sound of leaks and applying soap bubbles to suspicious locations. These methods have low inspection efficiency. In recent years, rapid progress in leakage detection methods have been made, including the mass balance method, flow model method, infrared thermography, optical fiber method and acoustic method [4,5,6,7,8,9,10]. These methods have their own advantages and disadvantages in terms of detection accuracy, efficiency, installation mode and cost.

For the mass balance method, monitoring the quantity of the fluid flowing through the inlet and outlet of the pipeline may be undertaken to determine whether leakage is occurring [11]. The principle of this method is relatively straightforward at the expense of its efficacy and reliability, because the change caused by a suspected leak can only be reflected after a period of time, especially for small flow leakage. Furthermore, more variables and parameters, such as fluid viscosity, pipe resistance, etc. need to be taken into account in the flow model. As such, a real-time flow model of the pipeline can be developed, which yields more reliable results compared to the mass balance method. Infrared thermography is mainly suitable and successful for leakage detection in pipe networks with obvious temperature characteristics, such as heating pipes [12]. Due to high temperature resolution, it has proven to be sensitive to small temperature differences and hence effective for detecting small leakages, which greatly improves the applicability of the method. More recently, optical fiber sensing technology has been developed for the detection of the temperature changes, vibration and strain caused by leakage in the pipe system [13]. To date, the optical fiber method has shown great promise due to its detection accuracy and efficacy. However, there are certain difficulties in installation for the pipelines buried under the soil or with wrapping materials.

Acoustic methods are commonly used for leakage detection due to the reliability, low cost and easy installation, including the traditional listening method, the negative pressure wave method and the acoustic emission method [14,15,16]. In principle, when leakage occurs, acoustic signals are generated in the vicinity of the leak, which are often seen as acoustic emission signals. The acoustic energy is transmitted in the pipe system in a variety of modes in broadband frequencies, which can be measured by acoustic and vibration sensors. Correspondingly, the leakage can be detected and located by analyzing the transmitted signals that carry the information of leak sources [17].

Acoustic emission based on cross-correlation is robust and widely used for pipeline leakage detection. The basic cross-correlation (BCC) process suffers from essential drawbacks, i.e., although this method is generally effective for white noise, the detection performance suffers to some degree for other-colored noise. To overcome this problem, an appropriate windowing or frequency weight function is introduced in the generalized cross-correlation (GCC) methods to pre-whiten the measured signals before cross-correlation. Nevertheless, the performance of the GCC methods varies in leak detection surveys due to the uncertainties in the practical engineering environments. Of particular interest is the phase transform (PHAT) cross-correlation, which uses only the phase information directly related to the difference between the arrival times of the leak noise at the sensors locations [18,19]. This method has proven to be effective for leak detection in a high signal-to-noise (SNR) environment. Otherwise, the detection accuracy suffers, in that the effects of background noise are neglected in the pre-whitening process. In recent years, some new time delay estimation (TDE) algorithms have been proposed. Ji et al. proposed a PHAT-β GCC algorithm by changing the exponential regulator β of the pre-whitening weight function in the 3D localization of transformer patrol robot [20]. Cui et al. proposed a variable step normalized LMS adaptive filter for leak localization in water-filled plastic pipes [21]. The algorithm transforms the TDE into the parameter estimation of the filter with advantages of variable step iterative learning. Alternative signal processing processes have been attempted to improve the leak detection accuracy [22,23,24,25,26], such as wavelet analysis, empirical model decomposition, neural network and deep learning methods. Whilst the advantages of these processes are immediately apparent, they also bring additional problems, such as large number of detection samples, long training time and complex calculated model.

In this paper, a secondary phase transform (PHAT) cross-correlation method is proposed to accentuate the information directly related to leakage. The method can suppress effectively the interfering effects of background noise by calculating the secondary cross-correlation function before the PHAT operation. Additionally, a sinc interpolation method is introduced for automatic searching the peak value of the secondary PHAT cross-correlation function. This can further improve the TDE accuracy since the error caused in the selection of the maximum value is significantly reduced. The remainder of this paper is as follows. In Section 2, the background of pipeline leakage detection is briefly discussed, followed by a detailed description of the proposed secondary phase transform (PHAT) cross-correlation. In Section 3 a numerical model is described, and some results are presented to show the detectability of the method for pipeline leakage detection. In Section 4 experimental work is carried out to verify the effectiveness of the proposed method in comparison with the BCC. Finally, conclusions are drawn in Section 5.

## 2. Methodology

### 2.1. Background of Pipeline Leak Detection

Figure 1 depicts the schematic diagram of the process of pipeline leak detection. The pipeline leakage will cause the fluid to overflow under the action of pressure and produce acoustic signals, which propagate along the pipeline upward and downstream. Acoustic/vibration sensors are generally installed at the pipe fittings, for example values and fire hydrants, to capture the transmitted leak signals.

Assume that the acoustic signals *x*_1_(*t*) and *x*_2_(*t*) received by sensor 1 and sensor 2 are expressed as follows:(1)$$\begin{array}{l} x_{1}(t)=s(t)+n_{1}(t)\\ x_{2}(t)=s(t-D)+n_{2}(t) \end{array}$$
where *s*(*t*) and *s*(*t*-*D*) represent the propagating leakage signals that travel along the pipe and are captured by sensors 1 and 2; the time delay *D* denotes the difference in the arrival times of the corresponding leakage signals at the sensors’ locations; and *n*_1_(*t*) and *n*_2_(*t*) denote the background noise at two sensor locations.

The BCC between the acoustic signals, *x*_1_(*t*) and *x*_2_(*t*), can be obtained by
(2)$$\begin{array}{l} R_{12}(\tau )=E[x_{1}(t)x_{2}(t-\tau )]\\ =E\{[s(t)+n_{1}(t)][s(t-D-\tau )+n_{2}(t-\tau )\}\\ {=R}_{\mathrm{ss}}(\tau -D{)+R}_{{\mathrm{sn}}_{2}}(\tau {)+R}_{{\mathrm{sn}}_{1}}(\tau -D{)+R}_{n_{1}n_{2}}(\tau ) \end{array}$$
where R_12_( ) denotes the BCC algorithm; R_ss_( ) is the BCC function between leakage signals *s*(*t*) and *s*(*t*-*D*); R_sn1_( ) and R_sn2_( ) are the BCC functions between the leakage signal *s*(*t*) and background noise signals *n*_1_(*t*) and *n*_2_(*t*) respectively; and R_n1n2_( ) is the BCC function between background noise *n*_1_(*t*) and *n*_2_(*t*). Generally, for random white noise, leakage signal and background noise signals are independent and uncorrelated with each other. In this case, Rsn_1_( ), R_sn2_( ) and R_n1n2_( ) in Equation (1) are equal to 0.

The above analysis suggests that if there is no leakage in the pipeline, the acoustic signals received by the two sensors are background noise, resulting in the cross-correlation result being close to zero. In contrast, when leakage occurs in the pipeline, the acoustic signals received by the two sensors include the additional transmitted leakage signals, which leads to the correlation result being non-trivial. This is further demonstrated in the simulations in the next section.

Applicable leak detection process involves two phases. When pipeline leakage is initially identified, background noise may produce signals with characteristics similar to leakage signals. Thus, it is critical to take accurate measurements to localize the exact location of the leak. Equation (2) can be further simplified as
(3)$$R_{12}(\tau )=E[s(t)s(t-D-\tau )]=R_{\mathrm{ss}}(\tau -D)$$

It is apparent that when *τ* − *D* = 0, the BCC function R_12_(*τ*) achieves the maximum value. In this case, the time delay between the two leakage signals *τ* = *D*. The time delay *D* is subsequently obtained by automatic searching the cross-correlation function R_12_(*τ*) for its peak value by using the sinc-interpolation. This can avoid the error caused by manually selecting the maximum value.
(4)$$D=\arg\max[\sum_{\tau =1}^{T}R_{12}(\tau )\mathrm{Sinc}(t-\tau )]$$
where argmax denotes the argument corresponding to the maximum value of the function; *T* is the signal length and Sinc denotes the sinc-interpolation.

Referring to Figure 1, given that the distance between sensor 1 and sensor 2 is *L*, the propagation wavespeed is *v*, and the time delay is *D*, the relative location from sensor 1 to the leak source, x can be obtained by
(5)$$x=\frac{L-vD}{2}$$

In the BCC method, the time delay is estimated by searching the peak of the cross-correlation function. This method has low computational complexity and is easy to be programmed. Accurate TDE can be achieved provided that leakage signals and background noise are uncorrelated with each other. However, when the uncorrelated assumption is violated in actual leak detection surveys, the TDE obtained by the BCC method will be in significant error due to the smearing effects on the main peak or even false peaks.

### 2.2. Secondary PHAT Cross-Correlation

In order to suppress the interfering effects of background noise, this paper proposes a secondary PHAT generalized cross-correlation method. The process of the proposed algorithm is shown in Figure 2, with detailed description as follows:

Step 1: calculate the auto-correlation function R_11_(*τ*) by performing auto-correlation operation on sensor signal *x*_1_(*t*)
(6)$$R_{11}(\tau )=E[x_{1}(t)x_{1}(t-\tau )]=E\{[s(t)+n_{1}(t)][s(t-\tau )+n_{1}(t-\tau )]$$

Step 2: calculate the cross-correlation function R_12_(*τ*) by performing the cross-correlation operation on sensor signals *x*_1_(*t*) and *x*_2_(*t*)
(7)$$R_{12}(\tau )=E[x_{1}(t)x_{2}(t-\tau )]=E\{[s(t)+n_{1}(t)][s(t-D-\tau )+n_{2}(t-\tau )]$$

Step 3: calculate the secondary cross-correlation function R_RR_(*τ*) by performing cross-correlation operation on the above auto-correlation function R_11_(*τ*) and cross-correlation function R_12_(*τ*)
(8)$$R_{\mathrm{RR}}(\tau )=E[R_{11}(t)R_{12}(t-\tau )]=E\{[R_{\mathrm{ss}}(t)+R_{{\mathrm{sn}}_{1}}(t)+R_{n_{1}s}(t)+R_{n_{1}n_{1}}(t)]$$

Step 4: the phase transform operation is used to weight the cross-power spectral density of R_RR_(*τ*) in the frequency domain. Finally, the secondary PHAT cross-correlation function, $R_{S\text{-}\mathrm{PHAT}}(\tau )$, is obtained by performing the inverse Fourier transform
(9)$$R_{S\text{-}\mathrm{PHAT}}(\tau )=\int_{-\infty }^{\infty }G_{RR}(\omega )\varphi_{12}(\omega )e^{-j\omega \tau }d\omega$$
(10)$$\varphi_{12}(\omega )=\frac{1}{\left|G_{RR}(\omega )\right|}$$
where *φ*12(*ω*) is the weighting function of the secondary PHAT cross-correlation in the frequency domain.

## 3. Simulations

Simulation results are presented for comparing the performance of the proposed secondary PHAT cross-correlation and the BCC method for leak identification and localization.

### 3.1. Leak Identification

#### 3.1.1. Pipe without Leakage

In the simulation model, white noise is used to simulate the signals received by the sensors when there is no leakage. Figure 3 plots the sensor signals in the time and frequency domains. The BCC function between the two sensor signals is calculated and shown in Figure 4. As can be seen from Figure 4, the BCC result value is very small. As anticipated in Section 2, it confirms that no leakage is occurring in the pipeline.

#### 3.1.2. Pipe with Leakage

When leakage occurs, the signals received by sensor 1 and sensor 2 are from the same leakage point with different arrival times. For simplicity, in the analysis, the sensor signal 1 is set to be the delayed signal of sensor 2 with 10 sampling points. The time domain and frequency domain characteristics of two sensor signals 1 and 2 are shown in Figure 5a,b, respectively. The corresponding BCC function is plotted in Figure 6. As expected, a distinct peak is found in the cross-correlation result. Further check of the BCC results for the pipes with and without leakage shows the effectiveness of the cross-correlation method for pipe leakage identification.

### 3.2. Leak Localization

Evaluation of the performance of TDE using the BCC and the secondary PHAT cross-correlation methods are performed under a high SNR (SNR = 10 dB) and a low SNR (SNR = −10 dB). Figure 7 plots the TDE curves of the two methods. The oscillatory behavior of the BCC result is obviously shown in Figure 7a,c, in particular, in the case of low SNR. As shown in Figure 7c, the fluctuations in the curve lead to a series of local maxima. However, these anomalous peaks are not generated by the leakage source; they are caused by noise interference instead. If these pseudo-peaks are mistakenly selected, the error in TDE will be significant. In contrast, the proposed secondary PHAT cross-correlation method leads to more reliable results for both high and low SNRs with single pronounced peaks corresponding to the actual time delay with some fluctuations in the magnitude being less distinctive than expected. This is due to the reason that the proposed method has the ability to pre-whiten the autocorrelation and cross-correlation, thus effectively reducing the interference effects of spurious peaks.

In order to better demonstrate the real peak corresponding to the actual time delay, Figure 8 plots the local results marked by the red boxes in Figure 7. For the BCC, the actual time delay is more discernible in Figure 8a, whereas it cannot be observed readily in Figure 7c. The simulation results in this section confirm that the proposed secondary PHAT cross-correlation method offers potential improvement for TDE over the BCC method.

## 4. Experiments

### 4.1. Experimental Setup

In order to evaluate the performance of the proposed method for pipeline leakage detection, experimental results from a PE water pipe rig are discussed. Tests were carried out at the leak detection facility built in the laboratory, as shown in Figure 9. The test pipe section had a length of 6 m and the wall thickness of 5 mm. The inlet and outlet of the pipe were, respectively, equipped with regulating valves. With reference to the figure, there was a simulated leak nozzle in the middle of the pipe section. Two hydrophones were used to capture the acoustic signals generated by the leak. The sensor model was a B&K 8103 hydrophone and the effective frequency range was 3 Hz–80 kHz. The distances between sensors 1 and 2 and the leakage was 0.9 m and 0.1 m, respectively. The leak signals were captured by using B&K PUSLE 3050 multi-channel signal acquisition instruments with a sampling rate of 8192 Hz. Similar to the analysis in the simulations, measurements were made in the cases of leak and no leak and in the pipe section.

The experimental procedures are briefly addressed as follows:

Step 1: Block the leak hole in the test pipe section to simulate the absence of leakage;

Step 2: Close the outlet valve and open the inlet valve of the test pipe section. Pressurize the test pipe by filling it with water. When the pressure reaches 1 bar and is stable, the sensors collect the acoustic signals in the pipe without leakage;

Step 3: Close the inlet valve and open the outlet valve of the test pipe section. Depressurize the test section. Then replace the nozzle with the leakage hole of 1 mm in diameter to simulate the pipeline leakage;

Step 4: Close the outlet valve and open the inlet valve of the test pipe section. Pressurize the test pipe by filling it with water. When the pressure reaches 1 bar and is stable, the sensors collect the leakage signals in the pipe.

### 4.2. Results and Discussions

When no leakage occurs, the signals measured by sensors 1 and 2 are plotted in Figure 10a,b in the time and frequency domains, respectively. In this circumstance, the signals include mainly ambient and system noise. Similar trends are demonstrated in Figure 10 for two sensor signals in both the time and frequency domains, indicating background noise dominates in the measured data. The noise signals have amplitudes of about 2.5 V, which is mainly concentrated at low frequencies below 50 Hz.

Next the cross-correlation methods are adopted to determine whether there exists leakage in the pipe section. Comparison of the BCC and the proposed secondary PHAT cross-correlation results for TDE curves is shown in Figure 11. Clearly there are a series of local peaks in the BCC results, as plotted in Figure 11a. In addition, these peak values are considerably larger, indicating that pipe leakage is likely to occur. This is, however, misleading, due to the fact that the BCC result is easily corrupted by the strong inference of background noise. In comparison, the TDE curve given by the secondary PHAT cross-correlation leads to very small result values in magnitude, which is close to 0 for the entire time-history. This confirms that the proposed method is marginally affected by background noise, and thereby can be adopted to monitor the pipe conditions more convincingly.

In the case of pipe leakage, the sensor signals are plotted in Figure 12. Compared to the signals without leakage plotted in Figure 10, the sensor signals have very different characteristics in both the time and frequency domains. Besides, it can be seen from Figure 12 that the signal at sensor 2 has larger amplitude compared to that at sensor 1 due to the closer distance relative to the leak source, and hence less attenuation. As shown in Figure 12b, most of the frequency components of the sensor signals are mainly concentrated in the frequency range up to 600 Hz. It must be noted that, similar to the case of no leakage, at lower frequencies below 50 Hz, the measured signals are dominated by background noise.

The cross-correlation methods are now applied to sensor signals for leak localization. Comparison of the TDE curves of the BCC and the proposed secondary PHAT cross-correlation are shown in Figure 13. Both methods are effective for leakage identification, since the main peak values of the BCC and the proposed methods are significantly large, confirming the presence of the pipe leakage.

As demonstrated in the simulations, several spurious peaks are found in the BCC results for test data as shown in Figure 13a. To better compare the TDE results, Figure 14 plots the local results marked by the red boxes around the main peak in [−0.05 s, 0.05 s] in Figure 13. The local result plotted in Figure 14a shows that the main peak has some fluctuations with close amplitudes. Again, they are caused by the inference of background noises at the test site. If the anomalous peaks are mistakenly selected as the peak corresponding to the time delay, an appreciable error will be given in the leak detection results. By comparison, the proposed secondary PHAT cross-correlation outperforms the BCC in terms of leak identification and localization. As stated above, the proposed methods via the pre-whitening process can effectively suppress the interference of background noise, hence producing more prominent peak corresponding to the actual time delay.

As can be found from the experimental setup, the distance difference between the two sensors is 0.8 m. In the measurements, the propagation wavespeed obtained is determined to be 310 m/s. As a result, the actual time delay is calculated to be 2.7 ms. It can be seen from Figure 14 that the BCC has several peaks around the actual time delay, while the proposed method leads to only one distinct peak. Their corresponding time delay results and calculation error are listed in Table 1. Compared to the BCC, the proposed secondary PHAT cross-correlation method is capable of suppressing other additional peaks unrelated to the time delay information.

## 5. Conclusions

In order to suppress the undesirable effects of background noise on pipeline leakage detection, a secondary PHAT cross-correlation has been proposed. The paper introduces the principle of the BCC and secondary PHAT cross-correlation methods for the process of pipeline leakage detection. Simulations have been conducted to compare the performance of the proposed method with the BCC method for leak identification and localization in the pipe, with and without leakage. It has been found from the cross-correlation curve that when there is no leakage, the correlation result is close to zero. However, when leakage occurs, the correlation curve will have an obvious peak value corresponding to the time delay resulting from the pipe leakage. The simulation results have shown that the proposed method has sharper peaks than the traditional BCC method in a low-SNR environment. Experiments were carried out on the pipeline test rig made in the laboratory. Test results have shown that the BCC method has multiple peaks, while the proposed secondary PHAT method has only one peak. Comparing the leakage detection error of different peaks, it can be seen that the proposed secondary PHAT method outperforms the traditional BCC method in terms of both the number of peaks and the average error. The findings suggest that the secondary PHAT cross-correlation method has been proposed as a potential solution to pipeline leakage detection in complex background environments, for example, in power plants or other industrial concerns.

## Figures and Tables

**Figure 1 sensors-23-01572-f001:**
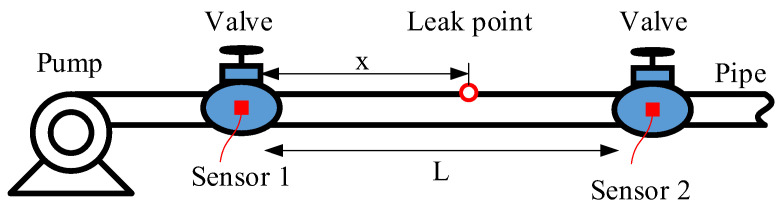
Schematic diagram of the process of pipeline leak detection.

**Figure 2 sensors-23-01572-f002:**
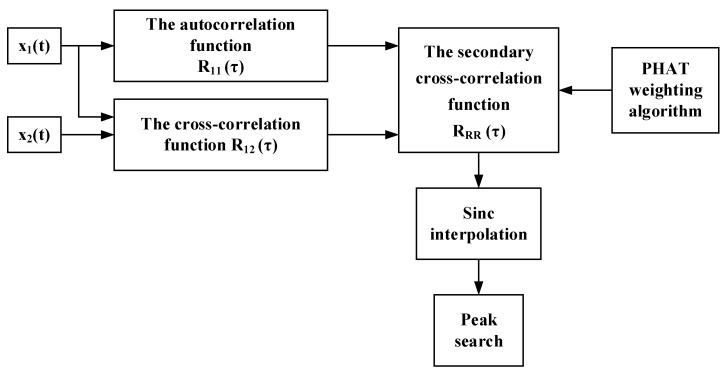
Schematic of the implementation of the secondary PHAT cross-correlation.

**Figure 3 sensors-23-01572-f003:**
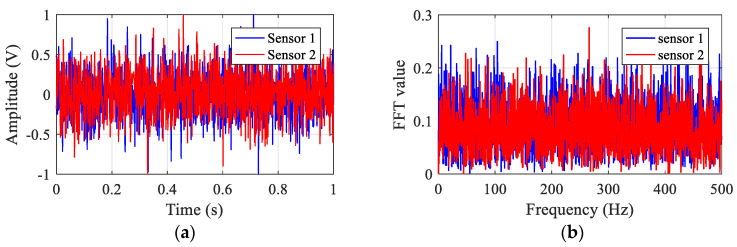
Simulated signals in the pipe without leakage: (**a**) in the time domain; (**b**) in the frequency domain.

**Figure 4 sensors-23-01572-f004:**
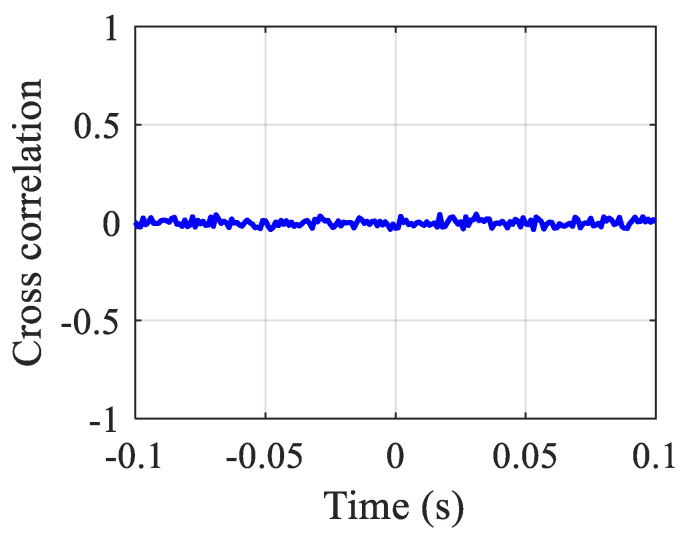
The BCC result of the simulated sensor signals for the pipe without leakage.

**Figure 5 sensors-23-01572-f005:**
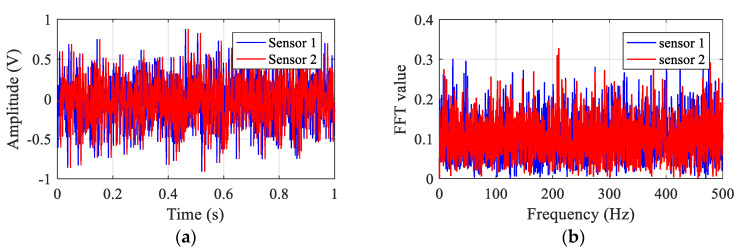
Simulated signals in the pipe with leakage: (**a**) in the time domain; (**b**) in the frequency domain.

**Figure 6 sensors-23-01572-f006:**
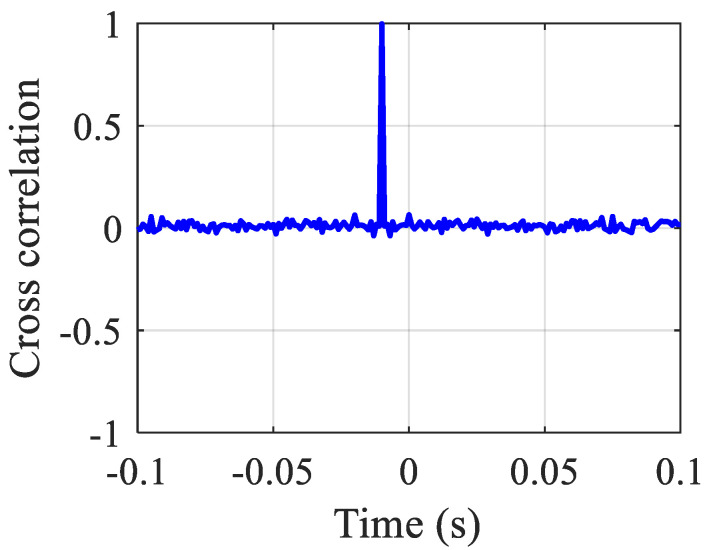
The BCC result of the simulated sensor signals for the pipe with leakage.

**Figure 7 sensors-23-01572-f007:**
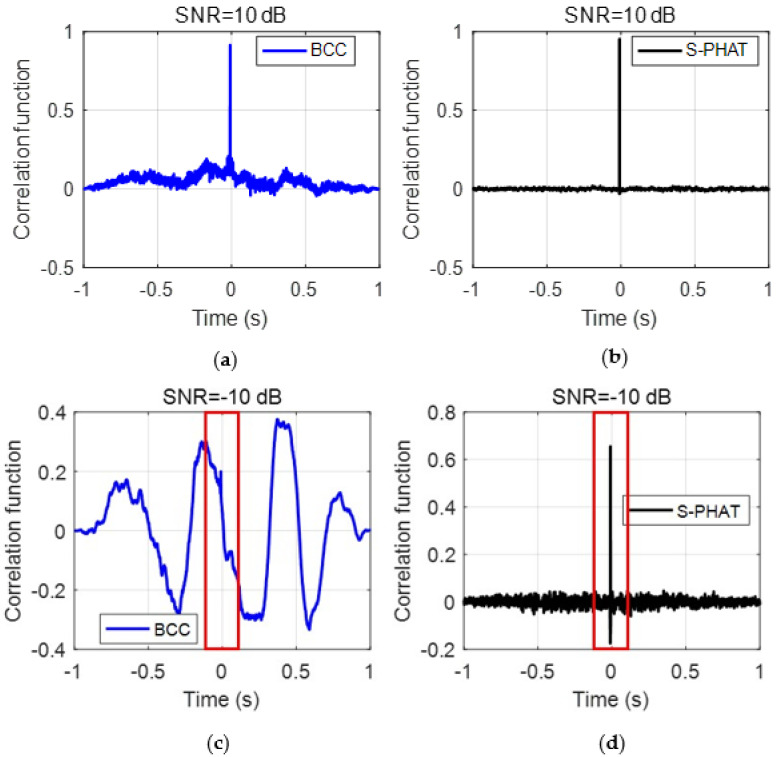
Comparison of the TDE curves of the simulated sensor signals: (**a**) the BCC method under SNR = 10 dB; (**b**) the secondary PHAT cross-correlation method under SNR = 10 dB; (**c**) the BCC method under SNR = −10 dB; and (**d**) the secondary PHAT cross-correlation method under SNR = −10 dB.

**Figure 8 sensors-23-01572-f008:**
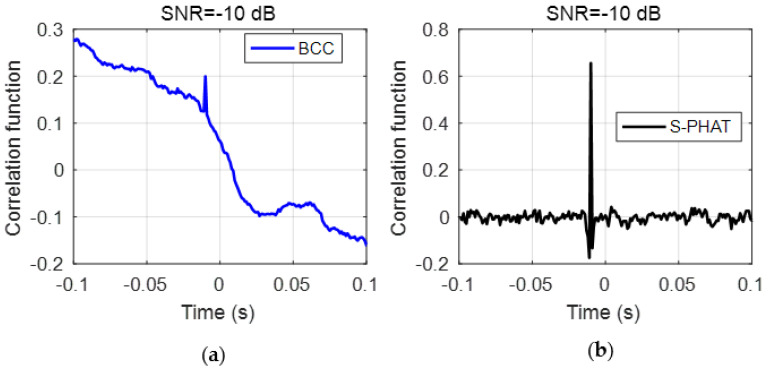
Local results marked by red box in Figure 7: (**a**) the BCC method under SNR = −10 dB; and (**b**) the secondary PHAT cross-correlation method under SNR = −10 dB.

**Figure 9 sensors-23-01572-f009:**
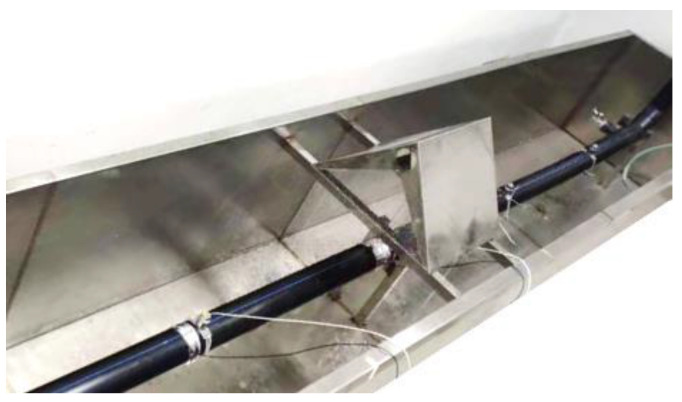
Experimental setup of the PE water pipe.

**Figure 10 sensors-23-01572-f010:**
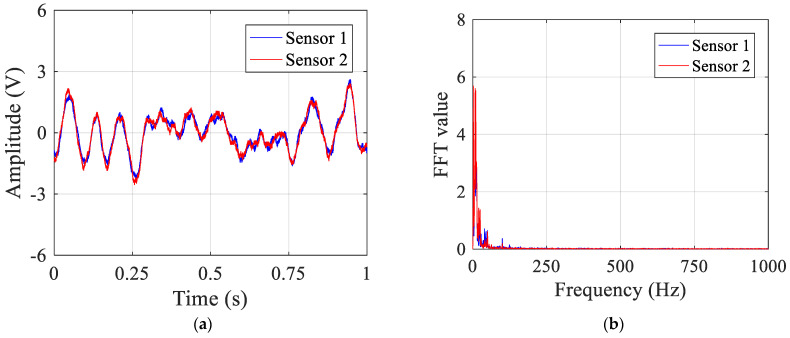
Test signals in the pipe without leakage: (**a**) in the time domain; and (**b**) in the frequency domain.

**Figure 11 sensors-23-01572-f011:**
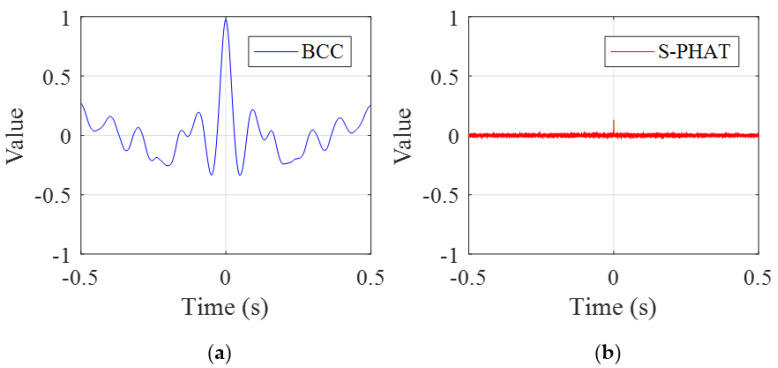
The TDE curves for the test pipe without leakage: (**a**) the BCC method; and (**b**) the secondary PHAT cross-correlation method.

**Figure 12 sensors-23-01572-f012:**
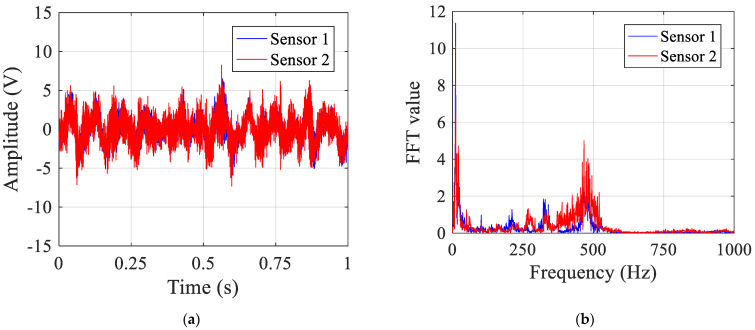
Test signals in the pipe with leakage: (**a**) in the time domain; and (**b**) in the frequency domain.

**Figure 13 sensors-23-01572-f013:**
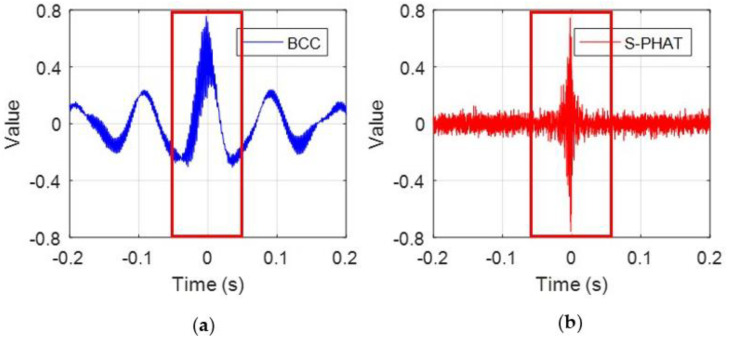
The TDE curves for the test pipe with leakage: (**a**) the BCC method; and (**b**) the secondary PHAT cross-correlation method.

**Figure 14 sensors-23-01572-f014:**
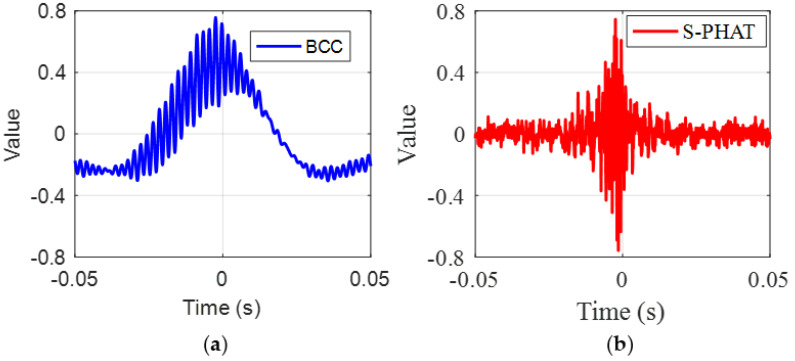
Local results marked by a red box in Figure 13: (**a**) the BCC method; and (**b**) the secondary PHAT cross-correlation method.

**Table 1 sensors-23-01572-t001:** Leak localization results.

Method	Peak	Real Time (ms)	Time Delay Estimation	Error (%)	Mean Error (%)
BCC	Peak 1	2.6	2.69	3.46	21.54
	Peak 2		2.56	1.54	
	Peak 3		2.44	6.15	
	Peak 4		2.32	10.77	
	Peak 5		0.37	85.77	
S-PHAT	Peak 1		2.44	6.15	6.15

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
