# Peer review of "Pipeline Leakage Detection Based on Secondary Phase Transform Cross-Correlation"

_sensors, 2023, doi:10.3390/s23031572_

Round 1

Reviewer 1 Report

Pipeline leakage monitoring is an important part for maintaining pipeline integrity. This paper presents a secondary phase transform(PHAT) cross-correlation method for leakage detection. It is verified that in the low SNR environment, secondary phase transform(PHAT) cross-correlation method is better than the basic cross-correlation (BCC) in noise suppression and time delay accuracy. Although this method has advantages over BCC method in low SNR environment, its universality and innovation could be improved.

Some other detailed comments are shown as follows:

1. In the introduction, only the traditional time delay estimation algorithm is mentioned, the latest time delay estimation algorithm proposed in the past five years should be given, such as PHAT- β Generalized cross-correlation algorithm for time delay estimation.

2. Secondary phase transform(PHAT) cross-correlation method has been studied and improved. The research gap and contribution of this paper should be pointed out more clearly in the introduction.

3. The secondary PHAT flow chart is too simple, and the peak detection method needs to be clear, such as fast Fourier transform (FFT), which will be more helpful for readers to understand.

4. In 3.2 leak location, methods can be compared in both high SNR and low SNR environments to increase the persuasiveness of the article.

5. In Section 3.1, other time delay estimation algorithms and methods can be introduced, such as phase transform generalized cross-correlation method (GCC-PHAT). BCC algorithm is very traditional, and some other methods can be used to compare.

6. The details of the experimental apparatus should be given.

7. The conclusions should be rewritten. 

Reviewer 2 Report

The tasks fo detection of leaks are not well addressed and convinced. The title, abstract, main body of the paper and conclusion are not well corresponded.

1. Sensor 1 is installed at the position of valve 1. How far is it from the pump? Will the position of sensor 1 affect the accuracy of leak location? How to consider this problem in this paper?

2. What is the specific model of the sensor in the experiment? No explanation in the paper?

3. Is the unit of horizontal and vertical coordinates in Figure 7 and Figure 8 missing?

4. How to eliminate the signal noise in Figure 3 and Figure 5? Whether this problem is considered in the paper?

5. This method is tested on 6m pipeline. Whether the engineering application is considered?

6. Does the method location formula in this paper still use the traditional location formula to find the cross correlation maximum? How to embody the innovation of the paper?

Reviewer 3 Report

An interesting technique that may be useful in a number of situations for cros correlating timeshifted continuous signals where there is semi-coherent noise.

- A couple of minor grammar errors in the introduction - other sections seem fine though

On page 10 would it be "fairer" to apply a simple highpass filter to the BCC result before comparing it? (I think your PHAT technique would still win out but anyone using BCC would do that I think).

Page 5 and 6 - should the time domain signals not be plotted with time on the x-axis rather than point number (not that is makes a huge difference as the graphs only show that you can't see anything by eye in the time domain!)

Otherwise great and clearly presented!

Round 2

Reviewer 2 Report

The author has revised it according to the comments and suggested adding some latest references, such as:

Q. Meng et al., "Leak Localization of Gas Pipeline Based on the Combination of EEMD and Cross-Spectrum Analysis," in IEEE Transactions on Instrumentation and Measurement, vol. 71, pp. 1-9, 2022, Art no. 9501209, doi: 10.1109/TIM.2021.3130680.
